# VideoDex: Learning Dexterity from Internet Videos

Kenneth Shaw*     Shikhar Bahl*     Deepak Pathak

Carnegie Mellon University

**Abstract:** To build general robotic agents that can operate in many environments, it is often imperative for the robot to collect experience in the real world. However, this is often not feasible due to safety, time, and hardware restrictions. We thus propose leveraging the next best thing as real-world experience: internet videos of humans using their hands. Visual priors, such as visual features, are often learned from videos, but we believe that more information from videos can be utilized as a stronger prior. We build a learning algorithm, VideoDex, that leverages *visual*, *action*, and *physical* priors from human video datasets to guide robot behavior. These actions and physical priors in the neural network dictate the typical human behavior for a particular robot task. We test our approach on a robot arm and dexterous hand-based system and show strong results on various manipulation tasks, outperforming various state-of-the-art methods. For videos and supplemental material visit our website at https://video-dex.github.io

**Keywords:** Dexterous Manipulation, Large Scale Robotics, Imitation Learning

## 1 Introduction

The long-standing dream of many roboticists is to see robots autonomously perform diverse tasks in diverse environments. To build a robot that can operate anywhere, many methods rely on successful robotic interaction data to train on. However, deploying inexperienced, real-world robots to collect experience may require constant supervision which is infeasible. This poses a chicken-and-egg problem for robot learning because to collect experience safely, the robot already needs to be experienced. How do we get around this deadlock?

Fortunately, there is plenty of real-world human interaction videos on the internet. This data can potentially help bootstrap robot learning by side-stepping the data collection-training loop. This insight of leveraging human videos to aid robotics is not new and has seen immense attention from the community at large [1, 2, 3]. However, most of the prior work tends to use human data as a mechanism for pretraining just the visual representation [4, 5, 6, 7, 8], much like how deep learning has been used as a pretraining tool in related areas of computer vision [9, 10] and natural language processing [11, 12]. Although pretraining visual representations can aid in efficiency, we believe that a large part of the inefficiency stems from very large action spaces. For continuous control, learning this is exponential in the number of actions and timesteps, and even more difficult for high degree-of-freedom robots (shown in Figure 1). Dexterous hands are one such class of high degree of freedom robots that have the possibility to provide great contact for the grasping and manipulation of different objects. Their similarity to human hands makes learning from human video advantageous.

In this work, we study how to go beyond using internet human videos merely as a source of visual pretraining (i.e. **visual priors**), and leverage the information of how humans move their limbs to guide train robots on how they should move (i.e. **action priors**). However, guiding robot motions using human videos requires understanding the scene in 3D, figuring out human intent, and transferring from human to robot embodiment. First, 3D human estimation works decently well in general human videos which we can leverage to gather 3D understanding. Second, there have been large-scale datasets that break down the human intent via crowdsourcing labels [2, 1]. Finally, to handle the embodiment transfer, we use human hand to robot hand retargeting as an energy function to pretrain the robot action policy. Our key insight is to combine these visual and action priors from human videos with a prior on how robot should move in the world [13, 14] (i.e., **physical prior**, using a second order dynamical system) to obtain dexterous robot policies that can act in the real world. We

---

*Equal contribution, order decided by coin flip.

6th Conference on Robot Learning (CoRL 2022), Auckland, New Zealand.

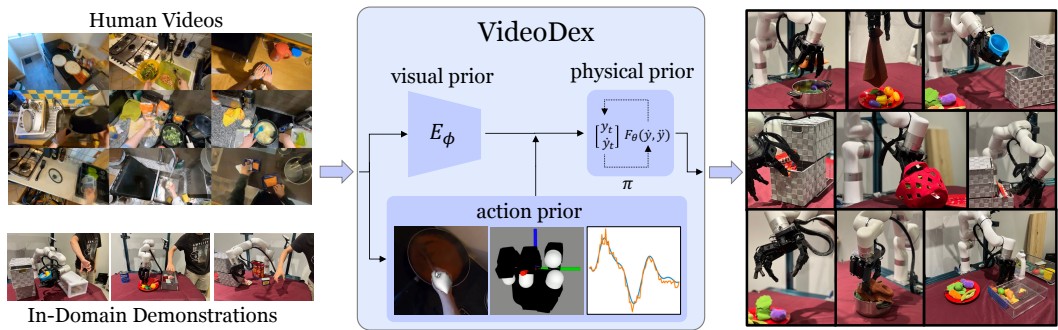

Figure 1: We re-target human videos as an action prior, use pretrainined embeddings as a visual prior, and use Neural Dynamical Policies (NDPs) [13] as a physical prior to complete many different tasks on a robotic hand.

call this approach, VideoDex. To enhance real-world performance, we mix the experience obtained from massive internet data with a few in-domain demonstrations.

In summary, VideoDex is a robot learning algorithm that incorporates visual, action, and physical priors into a single open-loop policy by learning from passive videos contained in human activity datasets from the internet. VideoDex then only needs to adapt to real world tasks using a few in-domain examples. We find that VideoDex outperforms many state-of-the-art robot learning methods on seven different real-world manipulation tasks on a high DOF multi-fingered robotic arm-hand system as well as on a 1-DOf gripper robotic arm system.

## 2 Related Work

**Learning for Dexterity** Reinforcement learning (RL) with an engineered reward function can show dexterous simulation results [15, 16] but requires lots of data, especially in high DOF dexterous manipulation. This requires simulators [17, 18], which cannot model physics properly, making real-world transfer difficult. Behavior cloning is an approach [19, 20] that can work safely. DIME [21] involves using nearest neighbor matching of image representations with demonstrations to determine actions. Qin et al. [22] teleoperates and learns policies in simulation, followed by Sim2Real transfer. DexMV[23] uses collected human hand videos for robot hand imitation learning. DexVIP [24] learns hand-object affordances and priors for RL initialization using curated video datasets.

**Learning from Videos and Large-Scale Datasets** There are many curated datasets from internet human videos, for example, FreiHand [25] for hand poses, 100 Days of Hands [26] for hand-object interactions, Something-Something [3] for semantically similar interactions, Human3.6M [27] and the CMU Mocap Database [28] for Human pose estimation. Epic Kitchens [2], ActivityNet datasets [29], or YouCook [30] are action-driven datasets we focus on for dexterous manipulation.

**Learning Action from Videos** Detecting humans, estimating poses of different body parts, or understanding the dynamics and interactions related to human motion is a commonly studied problem. One can model human hands using the MANO [31] model and the human body using SMPL, SMPL-X [32, 33] models. There are many efforts in human pose estimation such as [34, 35, 36]. We focus on FrankMocap [36] for our project as it is robust for online videos. Traditionally, teleoperation approaches have employed hand markers with gloves for motion capture [37] or VR settings [38]. Without gloves, Li et. al. [39] used depth images and a paired human-robot dataset for teleoperation, and Handa et. al. [40] designed a system that mimics the functional intent of the human operator to perform object manipulation tasks.

**Robot Learning by Watching Humans** Recent works have leveraged human datasets to learn cost functions [41, 42, 43], learn action correspondences [44] both in a paired [45] and unpaired manner [46]. This data can also be used to extract explicit actions by leveraging structure in the collection (such as reacher-grabber tools [47]) or prediction of future hand and object locations [48], as well as keypoint detectors [49]. This can also be used to build representations for robot learning [6, 50]. R3M [6] trains on the Ego4D [1] dataset using a temporal alignment loss between language labels and video frames. We build on top of previous efforts in this area, where we combine visual representations trained on human activity data, with *action* driven representations.

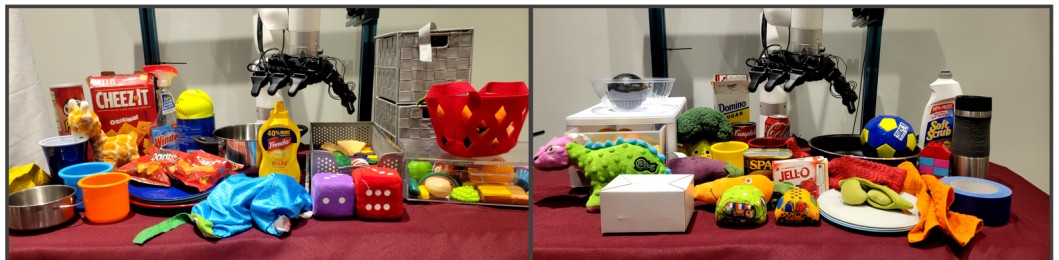

Figure 2: The collection of train objects (left) and test objects (right) used for experimentation.

# 3 Background

## 3.1 Neural Dynamic Policies

Neural Dynamic Policies (NDPs) [13, 14, 51], produce smooth and safe open-loop trajectories. When using them as a network backbone, they can be rolled out to trajectories of arbitrary lengths which enables the use of varying-length human videos. NDPs can be described with the Dynamic Movement Primitive equation [52, 53, 54, 55]:

$$\ddot{y} = \alpha(\beta(g - y) - \dot{y}) + f_w(x, g), \tag{1}$$

where $y$ is the coordinate frame of the robot, $g$ is the desired goal in the given coordinate frame, $f_w$ is a radial basis forcing function, $x$ is a time variable, and $\alpha, \beta$ are global constants. NDPs use the robot state, scene, and a NN to output the goal $g$ and shape parameters $w$ of the forcing function $f_w$.

## 3.2 Learning from Watching Humans

Recently, Sivakumar et al. [56] introduced Robotic Telekinesis, a pipeline that teleoperates the Allegro Hand [57] using a single RGB camera. Leveraging work in monocular human hand and body pose estimation [36], hand and body modeling [31, 32, 33], and human internet data, Robotic Telekinesis real-time re-targets the human hand and body to the robot hand and arm. Due to its efficiency and ease of use, we leverage Sivakumar et al. [56]'s approach for demonstration collection.

We borrow the human hand to robot hand retargeting method from Robotic Telekinesis [56] that manually defines key vectors $v_i^h$ and $v_i^r$ between palms and fingertips on both the human and robot hand. They build an energy function $E_\pi$ which minimizes the distance between human hand poses $(\beta, \theta)$ and robot hand poses $q$. $c_i$ is a scale parameter. Therefore, the energy function is defined as:

$$E_\pi(\,(\beta_h, \theta_h),\ q\,) = \sum_{i=1}^{10} ||v_i^h - (c_i \cdot v_i^r)||_2^2 \tag{2}$$

Sivakumar et al. [56] train an MLP $H_R(.)$ to implicitly minimize this energy function in 2, conditioned on knowing human poses $(\beta, \theta)$. For more details, we refer the readers to Sivakumar et al. [56].

# 4 Learning Dexterity from Human Videos

We learn general-purpose manipulation by utilizing large-scale human hand action data as prior robot experience. We leverage not only visual priors of the scene's appearance but also leverage important aspects of the human hand's motion, intent, and interaction. To do this, we *re-target* the human video data to trajectories from the robot's embodiment and point of view. By pretraining policies with these human hand trajectories, we learn *action* priors on how the robot should behave. However, it's notoriously difficult to leverage these noisy human video detections. Therefore, we must also employ a policy with *physical* priors to learn smooth and robust policies that do not overfit to noise. We explain insights and our method used to leverage *action* priors in the sections below.

## 4.1 Visual Priors from Human Activity Data

Many previous works [6, 7, 8] have tackled visual priors and representations for robot learning. These networks often encode some form of semantic visual priors into the pretrained network from human video internet datasets. We use the encoder from Nair et al. [6] as a useful visual initialization for our policy. Nair et al. [6] is trained on a visual-language alignment as well as a temporal consistency loss. Our network takes human video frames and processes them using the publicly released ResNet18 [58] encoder, $E_\phi$ from R3M [6]. The output of this network is our visual representation for learning.

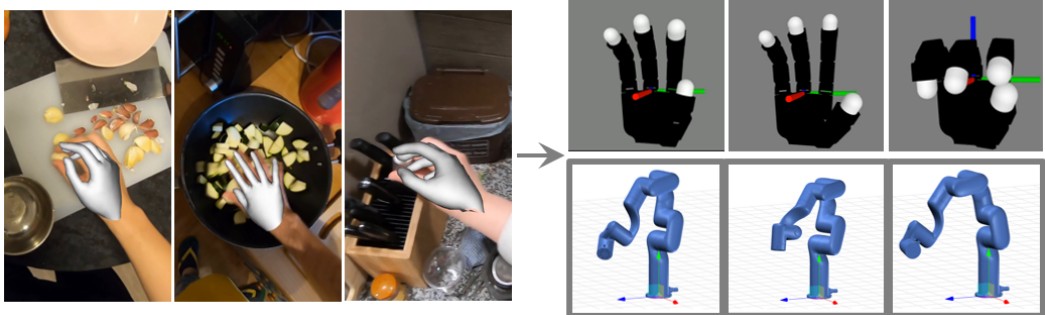

Figure 3: To use internet videos as pseudo-robot experience, we re-target human hand detections from the 3D MANO model [31] to 16 DoF robotic hand (LEAP) embodiment and we retarget the wrist from the moving camera to the xArm6 [59] embodiment. Videos at `https://video-dex.github.io`

## 4.2 Action Priors from Human Activity Data

While visual pretraining aids in semantic understanding, human data contains a lot more information about how to interact with the world. VideoDex uses action information to pretrain an action prior, a network initialization that encodes information about the typical actions for a particular task.

However, training robot policies on human actions are difficult, as there is a large embodiment gap between humans and robots as described in Handa et al. [40] and Sivakumar et al. [56] Thus, we must re-target the motion of the human to the robot embodiment to use it in training. This problem is solved using three main components. First, we detect human hands in videos. Second, we project hand poses $H$ to robot finger joints $H_r$. Finally, we convert human wrist pose $P$ to robot arm pose $P_r$. $H_r$ and $P_r$ define the trajectory of the human in the robot's frame, from which we can extract actions to pretrain our policy network with the action prior. See Figure 4 for a summary of the stages.

**Action and Hand Detections**   First, we must detect the right actions the human is completing. To expedite development, we use the action annotations from the EpicKitchens dataset [2] but an action detection network such as [60] can be used. Now, we must detect the hand. VideoDex first computes a crop $c$ around the operator's hand using OpenPose [61] and the result is passed to FrankMocap [36] to obtain hand shape ($\beta$) and pose parameters ($\theta$) of the 3D MANO model [31]. These parameters are passed through a low pass filter and subsequently used in re-targeting to the robot.

**Re-targeting Wrist Pose**   In this section, we show how to compute the transformation that describes the wrist pose in the robot frame denoted as $M_{Robot}^{Wrist}$. First, to calculate $M_{C_t}^{Wrist}$, where $C_t$ is the camera frame at timestep $t$ we leverage the Perspective-n-point algorithm [62]. This takes 2D keypoint outputs $(u_i, v_i)$ by the hand detection model and 3D keypoints from the hand model $(x_i, y_i, z_i)$ and computes $M_{C_t}^{Wrist}$. To accurately obtain camera intrinsics for PnP, COLMAP is used [63].

In human egocentric video datasets, the position of the camera is not fixed and we must compensate for this movement. Specifically, we compute the transformation between the camera pose in the first frame $C_1$ and all other frames in the trajectory, $C_t$. We call this transform $M_{C_1}^{C_t}$. To estimate this, we run monocular SLAM, specifically ORBSLAM3 [64].

Computing wrist poses in the first camera coordinate frame is important but this is still not in the robot frame because the robot is always upright. To be able to transform the human trajectory in the robot's frame, we must find the vector that is parallel to gravity in the camera's frame, $\alpha_p$. Thus recover object segmentations for surfaces that are parallel to the floor such as tables, floors, counters, and similar synonyms using a state-of-the-art object detector (Detic [65]). Then an estimated depth map from RGB frames only using Adabins [66] is computed. This way, the method does not rely on the long-term contiguity of a video like most SLAM approaches. We then use depth map portions that correspond to the relevant objects and calculate a surface normal vector. We estimate $\alpha_p$ using this normal vector and the following equations:

$$\text{pitch} = \tan^{-1}(x_{Acc}/\sqrt{y_{Acc}^2 + z_{Acc}^2}) \tag{3}$$

$$\text{roll} = \tan^{-1}(y_{Acc}/\sqrt{x_{Acc}^2 + z_{Acc}^2}) \tag{4}$$

Detailed ablations on the parameterization of the initial pitch of the predicted trajectory ($\alpha$) are provided in Section 6. In SLAM, we also remove the dependency on gyroscope data by assuming

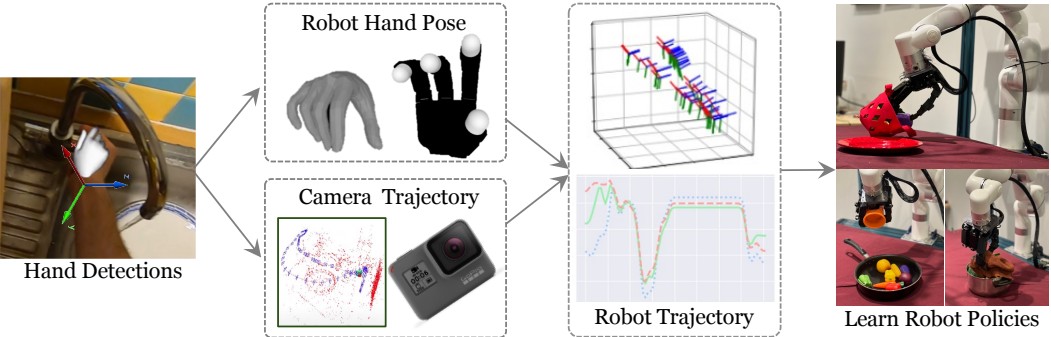

Figure 4: To use human videos as an action prior for training policies, we re-target them to the robot embodiment. The detected human fingers are converted to the robot fingers using a learned energy function. The wrist is re-targeted using the detections and camera trajectory and transformed to the robot arm.

that the scaling factor is 1.0. This is acceptable because the trajectory is rescaled to the robot frame later. Therefore, this wrist re-targeting approach uses only 2D images from human videos.

Since the robot has workspace limits, and we would also like to center the starting pose of the robot, we heuristically compute $T_{Robot}^{World}$ which rescales and rotates the human trajectory in the world frame $\tau_W^{\text{wrist}}$ into the robot trajectory $\tau_R^{\text{wrist}}$. The final function to obtain $M_{Robot}^{Wrist}$ can be described as:

$$M_{Robot}^{Wrist} = T_{Robot}^{World} \cdot M_{World}^{C_1} \cdot M_{C_1}^{C_t} \cdot M_{C_t}^{wrist} \tag{5}$$

---

**Algorithm 1** Procedure for VideoDex

---

**Require:** Human videos $V_{1:K}^H$ (length $T$), policy $\pi_\theta$, demonstrations $\mathcal{D}_{1:N}$. Human detection $f_{\text{human}}$ [36].
  **for** $k = 1...K$ **do**
    **for** $t = 1...T$ **do**
      Pose parameters $\theta_t, \beta_t = f_{\text{human}}(I_t)$
      Get wrist pose $w_t$ from 3, 4 and 5,
      Hand pose $h_t = H(\theta_t, \beta_t)$
    **end for**
    Store all $h_t, w_t$ into robot trajectory $\tau_R^k$
    $\hat{\tau}_R^k = \pi_\theta(I_1^k, h_1^k, w_1^k)$
    Optimize $\mathcal{L}_\theta = ||\tau_R^k - \hat{\tau}_R^k||_1$
  **end for**
  Store policy weights $\theta_h$ to initialize $\pi_\theta$
  **while** not converged **do**
    **for** $n = 1...N$ **do**
      $\tau_n, I_{1:T}^n = \mathcal{D}_n$
      $\hat{\tau}_n = \pi_\theta(I_1^n, h_1^n, w_1^n)$
      Optimize $\mathcal{L}_\theta = ||\tau_n - \hat{\tau}_n||_1$
    **end for**
  **end while**

---

Neural Dynamic Policies [13, 14].

**Re-targeting Hand Pose** Human hands are also in a different *embodiment* compared to that of robot hands, like our 16 DOF LEAP Hand [67] . Similarly, to Sivakumar et al. [56], we use $H(.)$ to map hand poses to robot hand poses. Given human detected pose $x_h$, we obtain $x_r = H(x_h)$ using a similar re-targeting network to Sivakumar et al. [56], and get human hand trajectories: $\tau_R^{\text{hand}}$ in the robot's embodiment. We use $\tau_R$ to denote the combined hand and wrist trajectories: $\tau_R^{\text{hand}}, \tau_R^{\text{wrist}}$. See Figure 3 for a visualization.

### 4.3 Learning with Human Videos

We must design an open-loop policy $\pi$ that learns first from the re-targeted human trajectories (the action prior) and then from real robot trajectories collected in teleoperation. Naively, training a neural network policy on $\tau_R$ will lead to overfitting to noisy hand detections. To circumvent this, we first use visual priors from the visual ResNet-based [58] encoder provided by Nair et al. [6], $E_\phi$. Then, we introduce a *physical prior* to the network, the physically-inspired

We construct $\pi$ with the following setup. We first process the first scene image $I$ with the visual encoder $E_\phi$. Then the extracted features $E_\phi(I)$ are used to condition an NDP for the wrist and hand separately, $f_{\text{wrist}}$ and $f_{\text{hand}}$. Concretely, each NDP operates by processing the input features with a small MLP which outputs $w, g$ which are the trajectory shape and goal parameters. The forward integrator of the NDP outputs an open-loop trajectory for the hand and the wrist, $\hat{\tau}_R$. We use the following loss function:

$$\mathcal{L} = \sum_k \text{Loss}_{L1}(\tau_R - [f_{\text{hand}}(E_\phi(I_k)), f_{\text{wrist}}(E_\phi(I_k))])$$

**Training Methodology:** We use between 500-3000 video clips of humans completing the same task category as the robot will from the Epic Kitchens dataset [2]. For example, in pick, there are

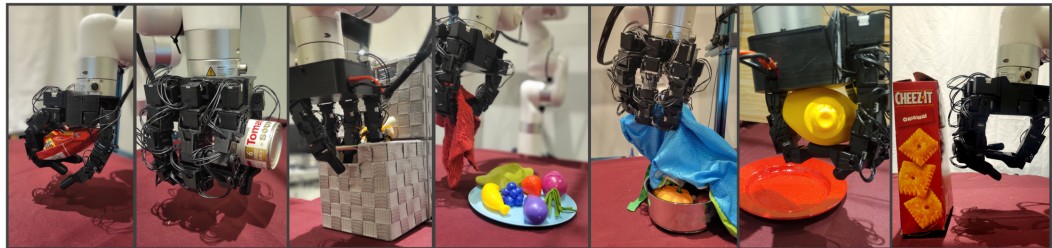

Figure 5: Tasks used in experiments. From left to right: pick, rotate, open, cover, uncover, place and push. See https://video-dex.github.io for videos of these tasks.

close to 3000 video clips of humans picking items. These are retargeted to the robot domain and used to pretrain the network with the human action prior of the pick task. Then, the final policy $\pi$ is trained on a few teleoperated demonstrations of pick on the real robot. The full training takes about 10 hours on a single 2080Ti GPU. More training details can be found in the appendix and in Algorithm 1. Our network consists of the R3M [6] initialized ResNet-18 [58]. We process these features with a 3-layer MLP with a hidden layer size of 512, which are then processed by 2 NDP [13] networks.

## 5 Experimental Setup

We perform thorough real world experiments on manipulation tasks, specifically many tasks that require dexterity. See our webpage for result videos. We aim to answer the following questions. (1) Is VideoDex able to perform general purpose open-loop manipulation? (2) How much does the action prior of VideoDex help? (3) How much does the physical prior of the NDPs in VideoDex help? (4) What important design choices are there (visual priors, physical priors, or training setup)?

**Task Setup** We pretrain action priors on retargeted Epic Kitchens data for seven robot tasks. Then, we collect about 120-175 demonstrations for each of these tasks on our setup to train the policy. In pick, the goal is to pickup an object. In rotate, the agent grasps and rotates the object in place. In cover and uncover, the goal is to cover or uncover a pan/plate with a soft cloth object. Push involves flicking/poking an object with the fingers. In place, the robot has to pick up an object and place it into a plate, pan or pot. In open we open three different drawers. Our testing procedure consists of unseen locations and objects. Details on the tasks and objects are in the supplemental.

While robot hands can provide great dexterity, we also investigate whether 2-finger grippers can benefit from action priors. The internet data is converted to where the closed human hand is a closed 2-finger gripper, and the open human hand is an open 2-finger gripper. We collect separate demonstrations on the real-robot using the 2-finger gripper from xArm [59]. Separate action priors are trained for the 16 DoF LEAP Hand and the 2-finger gripper.

## 6 Results

First, we evaluate the need for initialization with the action priors obtained from the human internet videos. $\theta_h$ The baseline without internet pre-training is called BC-NDP. It uses the same physical prior and visual network initialization, without the initialization from $\theta_h$. We also compare the effect of the action prior on 2-finger gripper policies. Second, we compare against two standard open-loop behavior cloning approaches introduced in recent benchmarks [51]. BC-open uses a 2 layer MLP instead of the NDP network. BC-RNN, uses an RNN to pre-process the visual features and then a two-stream, 2 layer MLP for wrist and hand trajectories. We try an offline RL ablation CQL [68], where we

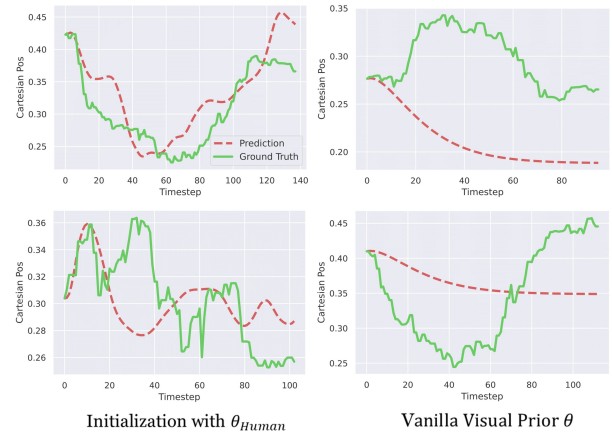

Figure 6: Networks initialized using action priors on human data without further training are closer to ground truth robot trajectories than networks only initialized using visual priors.

| | Pick | | Rotate | | Open | | Cover | | Uncover | | Place | | Push | |
|---|---|---|---|---|---|---|---|---|---|---|---|---|---|---|
| | train | test | train | test | train | test | train | test | train | test | train | test | train | test |
| BC-NDP [14] | 0.64 | 0.38 | **0.94** | 0.56 | **0.90** | 0.60 | **0.78** | 0.58 | 0.88 | 0.82 | 0.70 | 0.35 | 1.00 | 0.71 |
| BC-Open[51] | 0.50 | 0.44 | 0.72 | 0.38 | 0.80 | 0.40 | 0.44 | 0.58 | **1.00** | **0.91** | 0.40 | 0.25 | 1.00 | 0.93 |
| BC-RNN [51] | 0.56 | 0.31 | 0.78 | 0.50 | **0.90** | 0.50 | 0.56 | 0.42 | 0.88 | 0.75 | 0.70 | 0.50 | 1.00 | **1.00** |
| VideoDex | **0.83** | **0.77** | 0.85 | **0.71** | 0.80 | **0.80** | 0.75 | **0.63** | **0.96** | 0.92 | **0.89** | **0.80** | 1.00 | **1.00** |

Table 1: We present the results of train objects and test objects for Videodex and baselines as described above.

use the demonstrations as a sparse reward. We train a behavior cloning policy with the action prior from human videos without the physical prior of the NDP. We call this `VideoDex-BC-Open`. We ablate the type of visual representation and prior use by trying an initialization using the VGG16 network [69] (`VideoDex-VGG`) and the MVP network [7] [70] (`VideoDex-MVP`) based representation trained for robot learning. We ablate the need for a two stream policy, instead training a single NDP for both hand and wrist. (`VideoDex-Single`) To see if VideoDex works with fewer demonstrations (around 50 demonstrations, 5-7 per variant only), we train a policy called `VideoDex-Constrained`.

We analyze the results of our experiments and the guiding questions discussed in Section 5. We present the results of our findings as a 0-1 success rate in Table 1 and the result of the ablations we ran on the `place` task in Table 4.

**Effect of Action Priors** We firstly compare VideoDex against methods that do not employ an action prior trained on human data, as explained in Section 5. For almost all of the tasks, VideoDex either outperforms baselines or has a similar performance, especially for held out objects/instances. We believe that one of the key aspects of VideoDex generalizing to test objects is the action prior pretraining on human videos. This can be seen in Figure 6. Without ever training on the robot demonstrations, the trajectories initialized using the action prior pretrained network $\theta_h$ (left) are much closer to the ground truth trajectories of a network that is initialized using only a visual prior such as the encoder from Nair et al. [6] (right). From the results, we see that `VideoDex-BC-Open` with action priors (Table 4) outperforms `BC-Open`. Having a physical prior added (`BC-NDP`) tends to help, but it is not the case for every task. We suspect that some tasks require smoother behavior than others. Additionally, in Table 4 our offline RL baseline, `CQL` [68] does not perform as well as the rest of the approaches, even under-performing the Behavior Cloning setup. Qualitatively, we see a much less smooth and less safe execution with this method, thus we only perform it on one task (`place`). Note that we use the same visual prior for this as well.

| | Place | Open | Pick |
|---|---|---|---|
| 1-DOF BC-Open[51] | 0.62 | 0.69 | 0.71 |
| 1-DOF VideoDex | **0.69** | **0.82** | **0.77** |

Table 2: We compare how the 1-DOF xArm gripper performs using Videodex. [59] Separate demonstrations were collected using this gripper.

| | Place | Cover | Uncover |
|---|---|---|---|
| VideoDex-Fixed | 0.55 | 0.50 | 0.77 |
| VideoDex-Random | 0.45 | 0.63 | 0.85 |
| VideoDex-IMU | 0.70 | **0.67** | 0.90 |
| VideoDex | **0.80** | 0.63 | **0.92** |

Table 3: Ablations that compare the different ways of calculating the initial pitch of the camera with respect to gravity, on test objects. This enables us to transform human trajectories to be upright like the robot is.

**Hand vs 2-Finger Gripper** We compare whether the action priors from VideoDex also help in the more general 1-DOF gripper setting. In Table 2, we find that in the 1-DOF setting, VideoDex still improves performance on these tasks. This is because the priors from human internet videos still encode typical wrist trajectory behaviors as well as when the gripper should close for each task.

**Initial Pose Computation Comparison** We compare three different ways to estimate $\alpha_p$ or $M_{World}^{C_1}$, the vector that points parallel to gravity. These methods contrast with VideoDex which uses the surface normal of objects that are typically parallel with the floor to calculate the direction of gravity. `VideoDex-Fixed`, assumes that $\alpha_p$ is [0,0]. This is reasonable as we are not relying on robots to exactly mimic the human but get a general action prior. `VideoDex-Random`, randomizes $\alpha_p$ in the range of 15-45 degrees, which is the typical egocentric camera angle. `VideoDex-IMU` uses the internal image stabilization sensor data to estimate the upright vector. None of these approaches use gyroscope data in SLAM, as we assume that the scaling factor is 1.0. In Table 3, we present the results of these experiments. The performance degrades when randomizing or setting $M_{World}^{C_1}$ to a fixed value, in all three of the tasks, but it is still comparable to or better than our baselines that do not use any human action data. A possible explanation for the fact that

`VideoDex-Surface` performed better than our `VideoDex-IMU` is that the sensor data may be noisy and estimating surface normals from visual features is more robust.

**Effect of Physical Priors and Architectural Choices**   We compare different types of physical priors in Table 1 and in Table 4. In general (`BC-NDP`) tends to outperform baselines without a physical prior, except for `BC-RNN` in a couple of tasks. `BC-RNN` performs less aggressive behavior, which allowed it to efficiently grasp more objects. In Table 4 it's shown that an important physical prior is to treat the wrist and the hand in a more disentangled manner, as the performance for `VideoDex-Single` tends to drop compared to `BC-NDP` and `VideoDex-BC-Open` (Behavior Cloning with our action prior pretraining). The two stream architecture aids in learning, as it allows the policy to disentangle the actions of the wrist and the hand. This is important as the same grasp might be used for picking objects in many different locations, and similarly, it is possible to localize many objects and perform completely different types of interactions.

**Generalization with Less Data**   We limit VideoDex to a maximum of 5 and 10 teleoperated demonstrations per variant (we have 12-15 variants in our setup). As shown in Table 1, even with 5 instances per variant, we still see a 30% success rate for unseen objects. Empirically, the policies generally go to the right area but are not able to grasp objects properly. With less robot experience, VideoDex outperforms which demonstrates that action priors also boosts sample efficiency.

**Effect of Visual Priors**   We compared using our approach with MVP (`VideoDex-MVP`) [7] and VGG (`VideoDex-VGG`) [69] and their performance was below VideoDex using Nair et al. [6]. This is likely because both encoders are much larger than the ResNet18 [58] we use and require a lot more training time than feasible on human videos. However, `VideoDex-MVP` still performs better than `VideoDex-VGG`, which indicates that using a visual prior trained on human data does in fact help, as Xiao et al. [7] trained the representation in self-supervised fashion on videos and use the embeddings to perform robotics tasks in simulation. We see in Table 1, that while visual priors are important, action priors are more impactful.

|  | **Train** | **Test** |
|---|---|---|
| *Baselines*: |  |  |
| `BC-NDP` [14] | 0.70 | 0.35 |
| `BC-Open` [51] | 0.40 | 0.25 |
| `BC-RNN` [51] | 0.70 | 0.50 |
| `CQL` [68] | 0.40 | 0.20 |
| *No Physical Prior*: |  |  |
| `VideoDex-BC-Open` | 0.50 | 0.50 |
| `VideoDex-Single` | 0.50 | 0.30 |
| *Visual Prior Ablation*: |  |  |
| `VideoDex-VGG` | 0.20 | 0.20 |
| `VideoDex-MVP` | 0.40 | 0.20 |
| *Constrained Data*: |  |  |
| `VideoDex-Const-5` | 0.80 | 0.60 |
| `VideoDex-Const-10` | 0.50 | 0.30 |
| `VideoDex` **(ours)** | **0.90** | **0.70** |

Table 4: We present the results of the ablations discussed in Section 5. These are all performed on the `place` task.

**Choice of Robotic Hand**   In our experiments, we also tried using the Allegro Hand [57]. We found that the Allegro had higher inaccuracy in control and more hardware failures as compared to LEAP Hand. LEAP Hand outperformed the Allegro Hand $7 - 12\%$ on average in all experiments, thus we use it for our setup [67] .

## 7    Discussion and Limitations

Although we see strong results on the held-out objects, VideoDex has several limitations and scope for future work. First, we focus on curated human video datasets, such as EpicKitchens [2], but only use these as a convenience to expedite our process. It is possible to filter internet videos of humans according to tasks using action detectors and then processing them with VideoDex. We also use camera data in VideoDex but show that with a heuristic driven approach it is possible to obtain similar or better results. Second, we rely on off-the-shelf human hand detection modules that very often have erroneous 6D pose detections, especially when the hand is interacting with objects. Second, the action priors rely on the arm trajectory as well as the hand trajectory retargeting which must be recomputed for each different set of robot parameters and embodiment. Finally, our method of behavior cloning in the real world is currently open-loop, so it cannot react to changes in the environment. This is because closed-loop behavior cloning is difficult to keep safe in the real world. Similarly, when running closed-loop RL it is difficult to guarantee the safety of the system. We leave this to future work, to train policies that can react to changes in the real world.

**Acknowledgments**

We thank Aditya Kannan and Shivam Duggal for assisting in robot data collection. We thank Aravind Sivakumar, Russell Mendonca, Jianren Wang, and Sudeep Dasari for fruitful discussions. KS is supported by NSF Graduate Research Fellowship under Grant No. DGE2140739. The work is supported by Samsung GRO Research Award and ONR N00014-22-1-2096.

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
