# OpenReview forum: "VideoDex: Learning Dexterity from Internet Videos"
_robot-learning.org/CoRL/2022/Conference — CoRL 2022 Poster_

### Official Review · Reviewer_1pwQ · 2022-07-30

**Originality:** Good
**Technical Quality:** Good
**Clarity Of Presentation:** Fair
**Impact:** 3

**Recommendation:**

Weak Reject: I recommend rejecting the paper, but will not argue for my recommendation if the majority of other reviewers have a different opinion.

**Summary:**

VideoDex pretrains a policy network with videos, with gyroscope and accelerometer data, of humans performing a task, then fine-tunes with demonstrating trajectories collected by teleoperating the robot.  In order to train with the human data, they use the approach from [49] for mapping human pose to robot pose and use ORBSLAM3[55] to account for the camera motion.  They feed the image data, labeled with the outputted pose, into a ResNet18[15] backbone initialized with R3M's[6] features and use a Neural Dynamic Policy (NDP) [13] network to generate actions.  The paper demonstrates that using human data allows improved performance on 6/7 tasks.

**Issues:**

Line 47, Line 301 - The claim that the approach "learns from passive videos on the internet" is a little too strong.  Epic-Kitchens is a large and visually challenging source of data, but it is still curated and collected with a consistent methodology, making there be a significant gap between using it and using videos from the wilds of the internet.  Additionally, the proposed method makes use of gyroscope and accelerometer data for the human videos, which is not universally available for internet videos.

Line 84 - The related work section on learning manipulation from watching videos of humans only mentions a single recent paper.  There has been several years of work in this area by different groups that approach the problem in a variety of different ways. Here are some examples of other work that should be included and discussed.
Learning Robot Activities from First-Person Human Videos Using Convolutional Future Regression - https://arxiv.org/abs/1703.01040v2
Third-Person Visual Imitation Learning via Decoupled Hierarchical Controller - https://arxiv.org/abs/1911.09676
Learning Predictive Models From Observation and Interaction - https://arxiv.org/abs/1912.12773
AVID: Learning Multi-Stage Tasks via Pixel-Level Translation of Human Videos - https://arxiv.org/abs/1912.04443
Visual Imitation Made Easy - https://arxiv.org/abs/2008.04899
Model-Based Inverse Reinforcement Learning from Visual Demonstrations - https://arxiv.org/abs/2010.09034
Reinforcement Learning with Videos: Combining Offline Observations with Interaction - https://arxiv.org/abs/2011.06507
Learning by Watching: Physical Imitation of Manipulation Skills from Human Videos - https://arxiv.org/abs/2101.07241
Learning Generalizable Robotic Reward Functions from "In-The-Wild" Human Videos - https://arxiv.org/abs/2103.16817

Line 145 - Figure 4?

Line 179 - What are the differences from [49] that makes the retargetting network "similar" rather than just using the method from [49]?

Line 181 - Figure 3?

Line 218 - Should this be 120 demonstrations per task?  The appendix says 120-175 demonstrations per task and I'm not sure what the "hour" is measuring.

Table 1 - How many trials were the different policies evaluated on?  It would also be good to include some measure of variation in the results. That would help to show that the differences in performance are actually statistically meaningful.

Table 1 - What are the success criteria for the tasks?  For instance, the example for "Push" in the video looks more like the robot bumped into the object rather than executing a sustained pushing motion, but the table reports a 100% success rate.

Line 233 - Why is CQL not run on the rest of the tasks?

Line 289 - This experiment modifies both network architecture and network parameters.  A clearer experiment would be to compare ResNet18 with R3M features to ResNet18 with ImageNet features.  Also, the comparison between the performance with VGG and with MVP isn't fair, since VGG is an architecture that is 8 years out of date and should perform worse on pretty much any task, no matter the initialization.

Appendix - Which augmentations were used and which augmentations were not used?

One important experiment that I would like to see is the performance of VideoDex pretrained on human data from task A, then finetuned and tested on Task B.  If there was a significant gap between that and the performance when pretrained and tested on the same data, that would demonstrate that the human data is actually providing the model with useful, task-specific action priors rather than just putting it in a generally good initialization.


One of the central claims of the paper is that the use of NDP helps provide a physically grounded prior that enables the network to better deal with the noise in the trajectories/actions generated by [49]. It would interesting to compare the performance of VideoDex and VideoDex-Open pretrained with teleoperated demonstrations with different levels of noise artificially added.  If the performance gap increased as the level of noise increased, that would be convincing evidence that NDPs have some unique benefits in this setting.


**Quality Of The Limitations Section:**

Limitations are not well addressed

**Reviewer Expertise:**

4: The reviewer is confident but not absolutely certain that the evaluation is correct

**Robotics Focus:**

Sufficient demonstration on hardware

**Strengths And Weaknesses:**

Pros
The paper presents a theoretically simple method of learning from videos of humans.

The method is demonstrated on 7 different tasks, outperforming the baselines without human data on 6 of them.


Cons
The writing of the paper is somewhat scattered.

The analysis of why the proposed approach using NDP rather than a MLP works better with human data could be stronger.

The paper needs to be much clearer that it relies on gyroscope and accelerometer data from the human videos, which is a barrier to truly using internet-scale data.


**Summary Of Recommendation:**

I think this paper has promise, but could use some additional work to solidify its claims and clean up its presentation.

---

> ### Author Response · Authors · 2022-08-23
> **Response to Reviewer 1pwQ  (with new experiments) [Part 1/2]**
>
> Dear Reviewer,
>
> Thank you for the helpful comments and valuable feedback! We wanted to highlight that our approach does not necessarily rely on the camera's gyro/accelerometer data and we provide multiple ways to work purely from videos. We provide results and other clarifications below.
>
> We hope that these answers (+ new experiments) address all your concerns. If so, we kindly request the reviewer to consider updating the score. Otherwise, please let us know any further concerns that remain.
>
> > *Much clearer that it relies on gyroscope and accelerometer data from the human videos, which is a barrier to truly using internet-scale data.* and  *the proposed method makes use of gyroscope and accelerometer data… which is not universally available for internet videos.*
>
> - In our submitted version, we use the camera’s gyro/accelerometer data to transform the human camera pose to that of the robot's. This was more of a convenience than necessity as Epickitchen already had the camera data. However, our method does not rely on this camera data and can work purely from just videos.
> - Interestingly, we found that after removing the dependency on camera gyro/accelerometer data, the performance of VideoDex improved as it does not suffer from estimation error in camera’s gyro/accelerometer data. Please see the results and approach in the common comment above, or click here: [https://openreview.net/forum?id=qUhkhHw8Dz&noteId=pGj0s01CD71](https://openreview.net/forum?id=qUhkhHw8Dz&noteId=pGj0s01CD71)
>
>
> > *Epic-Kitchens ..is still curated and collected*
>
> - We can filter for videos that have some hand information as well as classify their actions to train for specific tasks, using off-the-shelf action detection and human detection models. We agree that EpicKitchens might be more curated, but we only used it as a convenience to expedite our process. We have clarified this in our paper.
>
>
> > *The related work section on learning manipulation from watching videos of humans only mentions a single recent paper.  Here are some examples……..*
>
> - Thank you very much for pointing these out, we have updated our related works section with a paragraph titled *Robot Learning by Watching Humans* to include works that we initially missed. These papers capture better the history of learning from human videos.
>
>
> > *writing of the paper is somewhat scattered.*
>
> - Thanks for pointing this out! We have made several edits to most sections of our paper in order to increase the clarity and convey our algorithm and approach better.
>
> > *compare ResNet18 with R3M features to ResNet18 with ImageNet features…Performance of VideoDex pretrained on human data from task A, then finetuned and tested on Task B….Performance of VideoDex and VideoDex-Open pretrained with teleoperated demonstrations with different levels of noise artificially added.*
>
> - We agree that these ablations are important and may provide additional insight. Firstly, we ran an ablation where we pertained a policy on human videos performing the place task and finetune it on the uncover task (using robot data). Similarly, we pretrained a policy on uncover and fine tuned on the place task. The results are in the below table under **VideoDex-Transfer**. We see that for both tasks the performance degrades slightly, especially in the place task.
>
> - We also train by adding noise to the demonstration trajectories, by adding two different levels of Gaussian noise with standard deviation being 0.01 and 0.05, shown as **VideoDex-Noise-0.01** and **VideoDex-Noise-0.05**. We find that adding more noise definitely hurts the performance of the method.
>
> - We also train ResNet18 features initialized from ImageNet classification features instead of the R3M features, and the results are under the column **VideoDex-ImageNet**. We can see that performance decreases, which indicates that the visual priors are important.
>
> - Note that all of the reported numbers are on test objects.
>
>
> |Task/Method   | VideoDex | VideoDex-Noise-0.01 | VideoDex-Noise-0.05| VideoDex-ImageNet|VideoDex-Transfer|
> |--------------|----------|---------------------|--------------------|------------------|-----------------|
> | Place        | 0.70      |0.55   |0.5    |0.40  |0.60|
> | Uncover     | 0.90    |0.87   |0.60  | 0.62 | 0.87|
>
>
> > *performance with VGG and with MVP isn't fair…VGG…is 8 years..worse on…any task*
>
> - We agree that a ResNet18 initialized with ImageNet features is a good ablation to run, and have provided the results for it, and have presented the results in the table above.
>
> > *Analysis of why the proposed approach using NDP rather than a MLP works better with human data could be stronger*
>
> - Humans often move in smooth manners, which can be efficiently captured by dynamical systems, thus our hypothesis is that NDPs would serve a better purpose than MLPs.
>
> ...part 2/2 continued below...

---

> > ### Author Response · Authors · 2022-08-23
> > **Response to Reviewer 1pwQ [Part 2/2]**
> >
> > >*Should this be 120 demonstrations per task? The appendix says 120-175 demonstrations per task and I'm not sure what the *hour* is measuring.*
> >
> > - We apologize for the confusion. 120-175 demonstrations for each task were collected, and this process takes about an hour per task.  We clarify this wording and add a table to the supplementary which enumerates the exact number of demonstrations that are used in training.
> >
> > |Task/Method   | VideoDex |
> > |--------------|----------|
> > | Pick         | 125     |
> > | Rotate     | 140     |
> > | Open       | 120     |
> > | Cover      | 124     |
> > | Uncover  | 145     |
> > | Place       | 175     |
> > | Push       | 136     |
> >
> > > *How many trials were the policies ..evaluated on..measure of variation in the results*
> >
> > - There were 24-30 trials performed for each task. We have updated the results table to include a measure of the variance.
> >
> > > *success criteria for the tasks…*Push* in the video looks more like the robot bumped*
> >
> > - The success was determined on a 0-1 binary scale from 2 human operators. It was determined from. For push, the object had to be moved at least an inch from its starting pose or have its orientation significantly changed.
> >
> > > *augmentations were used and which augmentations were not used*
> >
> > - We have updated the supplementary to share each augmentation and their details.  Specifically, we use Random Cropping from a scale of 0.8 to 1.0.  We use Random Grayscale with a probability of 0.05  We use Color Jitter with a brightness of 0.4, contrast of 0.3, saturation of 0.3 and hue of 0.3.  Finally, we normalize the RGB values to $\mu = (0.485, 0.456, 0.406)$,  $\sigma = (0.229, 0.224, 0.225)$.

---

> > > ### Author Response · Authors · 2022-08-27
> > > **Request for follow up (deadline today)**
> > >
> > > Dear Reviewer,
> > >
> > > Just gently checking once again to see if you had an opportunity to look at our rebuttal. We hope to have addressed all your concerns (including the major one about removing camera gyro/accelerometer data).
> > >
> > > Please let us know whether you have any further concerns remaining that prevent you from accepting the paper. The deadline for discussion is today.

---

> > > > ### Comment · Reviewer_1pwQ · 2022-08-27
> > > > **Response to comment**
> > > >
> > > > Thank you for your updates.  Could you also upload a copy of the updated pdf?

---

> > > > > ### Author Response · Authors · 2022-08-27
> > > > > **Follow up response: updated paper**
> > > > >
> > > > > Dear Reviewer,
> > > > >
> > > > > We have the paper ready with changes but it doesn't allow us to update the original paper above. But if we upload the anonymous file to this reply, it will immediately become public to everyone (note currently only abstract and reviews are public, not paper). PCs also sent this email few days ago:
> > > > >
> > > > > “**Q: Can I edit my original submission PDF?** A: Not during the rebuttal phase. If the paper is accepted, you will have to upload a final camera-ready version of the paper. Any uploads during the rebuttal phase will NOT change the original or final paper. ”
> > > > >
> > > > > All your comments strictly improved the results and quality of our method, so we promise that we have included them in the paper.
> > > > >
> > > > > However, if your acceptance is conditioned on reviewing the updated paper pdf, we will upload the anonymous pdf with a publicly visible link here. Please let us know.

---

> > > > > > ### Comment · Reviewer_1pwQ · 2022-08-27
> > > > > > **updated paper**
> > > > > >
> > > > > > It would be great if you could upload the anonymous pdf.  I think a lot of the concerns that I had about the initial submission were about overclaiming the results.  A lot of this has been addressed by the additional experiments, but it would be good to see how the writing was changed as well.

---

### Official Review · Reviewer_nwhC · 2022-08-01

**Originality:** Very Good
**Technical Quality:** Very Good
**Clarity Of Presentation:** Good
**Impact:** 4

**Recommendation:**

Weak Accept: I recommend accepting the paper, but will not argue for my recommendation if the majority of other reviewers have a different opinion.

**Summary:**

The authors demonstrate a system in which they combine a few different components to get interesting supervised-learned open loop behavior of real robot hands doing several different tasks. In particular the most notable part of the approach is using videos of human hands as an “action prior” which informs their supervised mapping.

**Issues:**

Please see the list in Weaknesses.

**Quality Of The Limitations Section:**

Limitations are addressed clearly

**Reviewer Expertise:**

5: The reviewer is absolutely certain that the evaluation is correct and very familiar with the relevant literature

**Robotics Focus:**

Sufficient demonstration on hardware

**Strengths And Weaknesses:**

# Strengths

- Good core idea. The overall idea of using action priors from human videos, via hand tracking, to make robots work better, is a good idea. There are a lot of closely related works, but I think they are well referenced in this paper.
- Good execution on several key parts. The execution details of handling moving cameras with camera pose tracking, together with per-frame hand tracking, seems to be well done. I also like just using R3M features out of the box, this is smart and interesting to see external validation.
- Results of real robots with hands doing a variety of things.

# Weaknesses

There are various unscientific elements of this paper in its current form.

While the work is interesting, I can’t recommend a strong accept for a paper in this form. Hopefully the list below will help the authors improve both this work and their future work.

If the authors can address all of the following weaknesses in their rebuttal, which I think is all doable and within scope to do in a rebuttal, I’d be happy to move from weak accept to strong accept.

1. It seems like the authors are not very upfront about the fact that this method does not produce closed loop policies. Only on the last page or two is it mentioned that the whole method is open loop. This is fine to study the task of (i) inputting an image of a scene and (ii) outputting an open loop trajectory, but, it of course is very limiting. The tasks are carefully chosen such that they don’t require any closed loop feedback. This aspect of their approach is not what most researchers in the field would expect… so a common experience of a researcher would be to look over the first handful of pages of this paper, and only at the last page or so realize that this is an open loop method. Please just make this clear up front.
2. Several false statements in the introduction:
  - “ To build such robotic agents that can operate anywhere, we need access to a lot of successful robot interaction data in many environments.” —> not necessarily true… This is a reasonable hypothesis, but one that isn’t tested in this paper, and it can’t be stated as a fact.
  - “ However, deploying inexperienced real world robots to collect experience must require constant supervision which is in feasible.” —> also not necessarily true… but also a very reasonable hypothesis. Just need to say “may require” instead.
  - “Most of the inefficiency in robot learning is due to the exponentially large action space.” —> an opinion, and can’t be stated as fact.
3. “NDPs can produce safe and smooth trajectories” … yes, but this is a meaningless statement. They *can* also produce trajectories that are completely unsafe. There is nothing about NDPs/DMPs that provides safety other than a bit of smoothness that may arguably help. But there is nothing that helps here with the presence of obstacles in the environment, or humans, etc. This statement probably only serves to confuse/mislead inexperienced readers, please remove/fix.
4. The paper mentions a “physical” prior as a key component, but this is just that it uses Dynamic Movement Primitives it seems. I’m not sure this is the best way to communicate this. Line 191 also says physically-aware NDPs… they don’t know anything about contact physics… maybe just say second order system or dynamical system or something, maybe physically-inspired, but not physically-aware. And whenever it says, for example line 269, “baselines without a physical prior” it should just be instead clear that this just means they don’t use DMPs.
5. Line 213 “ is VideoDex able to perform general purpose manipulation?” Since the method is open loop, the answer is no. That’s fine, and the results are still impressive, but should be clarified… this is not something that needs to be empirically evaluated, it’s just a result of the formulation.
6. It’s very confusing that citation 44 is used open loop… this isn’t an intention of the method. Also, is the RNN version closed loop over time? It’s not clear. And if it’s not? … I’m not sure how the RNN would be any different if it’s not used sequentially over time.
7. Please state exactly how many demonstrations were used for the different experiments.
8. In the conclusion… “ this is because training RL in the real world is difficult due to hardware limitations.” Yes, but this isn’t reason to make the used behavior cloning method open loop instead of closed loop.


## Minor

Don’t worry about these too much but I mention these as opportunities to improve the paper further.

- Ego4D is not cited on page 2 (mentioned but not cited)
- HR() is not defined in an equation. Also, I would recommend not using two letters for a math symbol… it looks like a matrix H multiplied by a matrix R
- Why use ORBSLAM3 rather than COLMAP for the poses? Already running colmap for the calibration.




**Summary Of Recommendation:**

Please see the start of the Weaknesses section where I explain the recommendation.

---

> ### Author Response · Authors · 2022-08-23
> **Response to Reviewer nwhC (with new experiments)**
>
> Dear Reviewer,
>
> Thank you for the helpful comments and valuable feedback! We have updated our writing to reflect your comments and provide clarifications below.
>
> We hope that these answers address all your concerns. If so, we kindly request the reviewer to consider updating the score. Otherwise, please let us know any further concerns that remain.
>
> > *not very upfront about the fact that this method does not produce closed loop policies…make clear upfront*
>
> - Thanks for pointing this out. We agree and have made it much clearer in our introduction and method section, as well as in the limitations.
>
> > *'is VideoDex able to perform general purpose manipulation?' Since the method is open loop, the answer is no.*
>
> - We have edited the paper and our experiments section to reflect this. The question asked is now:
> *Can VideoDex perform general purpose open-loop manipulation?*
>
> > *Several false statements in the introduction*
>
> - Thanks for pointing these out, we have addressed and clarified the language for all of these in our introduction.  We follow your advice and clearly state these as opinions.
>
> > *[NDPs/DMPs producing smooth and safe trajectories]...nothing that helps here with the presence of obstacles in the environment, or human*
>
> - We agree that NDPs or DMPs do not incorporate knowledge about obstacles or surrounding humans.  However, we have found that the smooth trajectories that are output are safer for the robot hardware. We have updated the wording in the paper.
>
> > *[say NDPs are] maybe physically-inspired, but not physically-aware.*
>
> - Thanks for pointing this out. We agree with this statement and have made this change in our paper.
>
>
> > *It’s very confusing that citation 44 is used open loop…RNN version closed loop over time?...not sure how the RNN would be any different if it’s not used sequentially over time.*
>
> - Similar to RB2 [44], we use the RNN policy to output a long sequence of trajectory points from a single image. Since this is a different trajectory parameterization than NDP,  we compared against it in a similar way as to how our NDP policy works.
>
> > *state exactly how many demonstrations were used for the different experiments.*
>
> - We show this table below and have added it to the supplementary:
>
> |Task/Method   | VideoDex |
> |--------------|----------|
> | Pick         | 125      |
> | Rotate       | 140      |
> | Open         | 120      |
> | Cover        | 124      |
> | Uncover    | 145      |
> | Place        | 175      |
> | Push         | 136      |
>
> > *‘training RL in the real world is difficult due to hardware limitations’... isn’t reason to make the used behavior cloning method open loop instead of closed loop*
>
> - Thanks for pointing this out! This was originally misworded and has been updated in the paper.  By using behavior cloning open loop, we can verify safety before running the policy.  Closed loop policies cannot be verified for safety before deployment.
>
> > *Why use ORBSLAM3 rather than COLMAP for the poses? Already running colmap for the calibration.*
>
> - The main reason for this is that we can ORBSLAM3 [55]  a lot faster than COLMAP [54]. If the source of each video was different, we would have to run COLMAP to estimate intrinsics every time, in which case COLMAP would be more efficient. However, in our case the source of the videos are from similar cameras, thus we only need to run COLMAP on a few of them.
>
> > *Ego4D is not cited on page 2 (mentioned but not cited)*
> > *HR() is not defined in an equation. Also, I would recommend not using two letters for a math symbol… it looks like a matrix H multiplied by a matrix R*
>
> - Thanks for pointing this out - we fixed this!
>
> ### Removing the dependency on camera gryo/accelerometer data
> All other reviewers and AC had another concern that the method needs camera gyro/accelerometer data but this is not the case. This was more of a convenience than necessity as EpicKitchens already had the camera data. However, our method does not rely on this camera data and can work purely from just videos. Interestingly, we found that after removing the dependency on camera gyro/accelerometer data, the performance of VideoDex improved as it does not suffer from estimation error in the camera’s gyro/accelerometer data. Please see the results and approach in the common comment above, or click here: [https://openreview.net/forum?id=qUhkhHw8Dz&noteId=pGj0s01CD71](https://openreview.net/forum?id=qUhkhHw8Dz&noteId=pGj0s01CD71)

---

### Official Review · Reviewer_KbvG · 2022-08-01

**Originality:** Good
**Technical Quality:** Good
**Clarity Of Presentation:** Good
**Impact:** 4

**Recommendation:**

Weak Accept: I recommend accepting the paper, but will not argue for my recommendation if the majority of other reviewers have a different opinion.

**Summary:**

In this paper, the author investigates how to utilize large-scale human video to train dexterous robot manipulation skills. To leverage the information from the Internet videos, the author proposes a handful of techniques to pre-process the video data to extract the action information. Then the network is trained on the extracted hand data and deployed to the real robot with some human demonstration collected by teleoperation for fine-tuning. Experiments show that the proposed pipeline can solve multiple manipulation tasks.


**Issues:**

- Clarification for the weakness part listed in the previous section.
- More explanation about the transformation used in Eq5:
  - How to heuristically compute $T_{Robot}^{World}$
  - How to compute the translation term for $M_{C_1}^{World}$
  - How to get the 3D keypoints used to compute $M_{C_t}^{Wrist}$ by PnP
- Link in supp is broken with 404 error: https://videodex.github.io/download
- Clarification on the benefits of using dexterous hand for the task defined in the paper compared to normal parallel gripper.



**Quality Of The Limitations Section:**

Limitations are addressed clearly

**Reviewer Expertise:**

4: The reviewer is confident but not absolutely certain that the evaluation is correct

**Robotics Focus:**

Sufficient demonstration on hardware

**Strengths And Weaknesses:**

**Strength**

- The direction explored in this paper is important. Utilizing the internet video data for robot learning is well motivated. Especially considering the similarity between human and multi-finger hands, this direction looks very promising.

- The authors perform experiments with multiple real-world tasks with pick and place, pushing, and rotating objects.

**Weakness**

- Although the objective of this paper is very impressive, the experiments can not support the introduction and there are multiple overclaims.

- Section 4 is titled VideoDex: Learning Dexterity from Youtube. However, I can not find any evidence that the author utilizes YouTube data for learning dexterous manipulation. As mentioned in the Section on Retargeting Wrist Pose, ORB SLAM and the camera’s acceleration data are used to compute the camera pose trajectory. This information is not readily available in the YouTube data. The experiments and methods are misaligned with this claim.

- In the introduction line 42, the author mentioned that our key insight is to combine these visual and action priors from passive data with the physical constraints of how robots should move in the world. However, the method does not consider the surroundings of the human hand, and the detection results itself is not accurate. How to incorporate physical information into the training data?

- Missing literature discussion on previous learning from video works:

    [1] *DexMV: Imitation Learning for Dexterous Manipulation from Human Videos, 2021*: This paper focuses also on how to learn dexterous manipulation from human videos. The reviewer understands that this literature paper uses simulated tasks while the authors focus on the real robot settings. But it seems that similar pipelines are also used in this paper: estimating the human hand, retargeting, and learning from retargeted hand pose.

    [2] *The Surprising Effectiveness of Representation Learning for Visual Imitation, 2021*: This paper also focuses on how to leverage the video data for better learning. It also uses a GoPro camera to collect a video of each trajectory, which is the same as the Ego4D dataset used in this paper. It shows that by learning from this video data, the final manipulation performance can be improved a lot.

    These literature works use very similar methods to achieve robot learning. The novelty claims of this paper can also be found in this literature.

- Missing details for Re-targeting Wrist Pose
The detection module FrankMocap is a 2D hand detector, it is not clear how the author can get 3D keypoints from the hand model in the camera frame. Also, this section is important in the whole technical approach, it is better to provide visualization of the final retargeted robot. A hand wrist pose and robot arm should also be visualized in Figure 3 if they are used in the training. If the wrist pose and arm joint pose is not used, how to pretrain the action prior?

- Missing details about transforms
In the equation, it is not clear why the author uses T and M to denote pose simultaneously. What are the differences? If M is also a $SE(3) $transformation, how to compute the position part of the $M_{World}^{C_1}$? Besides, the reviewer can not find any information about how the $T_{Robot}^{World}$ is determined heuristically in both the main paper and supplementary.


**Summary Of Recommendation:**

The paper studies an important problem of learning manipulation skills from YouTube videos. The objective is of vital importance to the field. However, the methods described in the paper are not able to support the claim. The manipulation tasks are diverse but lacking in technical difficulty, which can also be achieved by parallel gripper. More investigations are necessary.

## Updates (after rebuttal)

The updated version removes the over-cliam of learning from YouTube and utilize visual RGB-D slam techniques to learn from image only. The reviewer believes that this is an important direction in imitation learning on dexterous manipulation. Although the experiments of this work is not so solid, i.e. missing intermediate evaluation results of the pretraining data generated from videos, the overall technical pipelines sounds reasonable to me. The reviewer raise the score to weak accept under the promise of the author that:

Shows intermediate visualization/results of the generated data for pretraining. The current experiments only show the final learning performance of manipulation tasks. It is unknown that what is the quality of these generated data from videos and how the quality of these data will influence the final performance in a more strict way. It will be preferable if the author can show some visualization of the generated VideoDex data in a more straight-forward way. For example, visualize the robot hand-arm trajectory and compared with the original internet video.

---

> ### Author Response · Authors · 2022-08-23
> **Response to Reviewer KbvG (with new experiments)**
>
> Dear Reviewer,
>
> Thank you for the helpful comments and valuable feedback! We wanted to highlight that our approach does not necessarily rely on the camera's gyro/accelerometer data and we provide multiple ways to work purely from videos. We provide results and other clarifications below.
>
> We hope that these answers (+ new experiments) address all your concerns. If so, we kindly request the reviewer to consider updating the score. Otherwise, please let us know any further concerns that remain.
>
> > *the experiments can not support the introduction and there are multiple overclaims
> Retargeting Wrist Pose, (‘how to compute $M_{C_1}^{Wrist}$ ’), ORB SLAM and the camera’s acceleration data are used to compute the camera pose trajectory…information is not readily available in the YouTube*
>
> - In our submitted version, we use the camera’s gyro/accelerometer data to transform the human camera pose to that of the robot's. This was more of a convenience than necessity as Epickitchen already had the camera data. However, our method does not rely on this camera data and can work purely from just videos.
> - Interestingly, we found that after removing the dependency on camera gyro/accelerometer data, the performance of VideoDex improved as it does not suffer from estimation error in camera’s gyro/accelerometer data. Please see the results and approach in the common comment above, or click here: [https://openreview.net/forum?id=qUhkhHw8Dz&noteId=pGj0s01CD71](https://openreview.net/forum?id=qUhkhHw8Dz&noteId=pGj0s01CD71)
>
> > *Missing details for Re-targeting Wrist Pose…not clear how the author can get 3D keypoints from the hand model in the camera frame*
>
> - The wrist-retargeting does not need any extra information other than the RGB image. The FrankMocap [39] hand model detects paired 2D keypoints in the image and 3D keypoints in the SMPLX [36] canonical frame. We use the Perspective-n-Point approach to transform these 3D keypoints into the camera frame. COLMAP [54] is used (which only needs 2D images) to estimate camera intrinsics. All of the steps of our updated retargeting approach are obtainable from generic human videos without any additional data.
>
>
> > *Missing details about transforms…differences between $T$ and $M$, $T_{Robot}^{World}$  (see review for equations)   *Besides, the reviewer can not find any information about how the $T_{Robot}^{World}$  is determined heuristically in both the main paper and supplementary.*
>
> - Thanks for pointing this out. We firstly wanted to clarify that $M$ denotes a linear transformation matrix, whereas $T$ is a non-linear transformation.  We have updated the paper to clarify this distinction.
>
> - $T_{Robot}^{World}$ contains the transform between the canonical camera frame and the canonical robot frame with a roll, pitch, yaw of [$-\frac{\pi}{2},0 ,-\frac{\pi}{2}$]. We also rescale the trajectory in correspondence to the human workspace limits (randomizing the scaling factor by +-10%). We clarify the wording of this in the supplementary.
>
> - $M_{C_1}^{World}$ does not contain a translational component, only a rotational component to make the trajectory upright like the robot is.
>
> - To obtain the 3D keypoints, we run the FrankMocap [39] hand detector. The output of this model contains 3D positions for different hand joints in the SMPLX [36] canonical frame. We then transform these 3D keypoints to be relative to the wrist joint. Using corresponding 2D keypoints, we apply the PnP approach to compute the transform between the camera and the wrist. We have clarified details in the paper.
>
>
> > *method does not consider the surroundings of the human hand…How to incorporate physical information into the training data*
>
> - Currently we implicitly consider the surrounds of the human hand as we use a visual policy to output actions. It is possible to make this more explicit by adding in the output of an object detection module, which we leave for future work.
>
> > *hand wrist pose and robot arm should also be visualized in Figure 3…provide visualization of the final retargeted robot*
>
> - Thank you for this comment!  We agree with you that it is better to show the full robot retargeting instead of just the hand and have thus edited Figure 3. It can be seen [here](https://videodex.github.io/Figure_3.pdf).
>
> > *benefits of using dexterous hand for the task defined..compared to normal parallel gripper.*
>
> - We believe that dexterous hands can provide more contact for grasping and manipulating objects. They are also of a similar embodiment as human videos, therefore we can extract more information from passive videos. We have updated the second paragraph of our introduction to motivate this point better.
>
> > *Missing citations: DexMV….The Surprising Effectiveness of Representation Learning for Visual Imitation*
>
> - Thanks for pointing this out. We had cited and discussed DexMV in the Related Works, and we have now added discussion of Pari et al., 2021.

---

> > ### Author Response · Authors · 2022-08-27
> > **Request for follow up (deadline today)**
> >
> > Dear Reviewer,
> >
> > Just gently checking once again to see if you had an opportunity to look at our rebuttal. We hope to have addressed all your concerns (including the major one about removing camera gyro/accelerometer data).
> >
> > Please let us know whether you have any further concerns remaining that prevent you from accepting the paper. The deadline for discussion is today.

---

> > > ### Comment · Reviewer_KbvG · 2022-08-27
> > > **Response to Author Response**
> > >
> > > Dear Authors,
> > >
> > > Thanks for effort for the new experiments and appreciate your detailed response. I will listed my response to response one by one:
> > >
> > > 1. `including the major one about removing camera gyro/accelerometer data`: In my original response, I think it is okay to use gyro/accelerometer data as long as the authors can do more accurate claim in the paper. If the experiments are not performed on YouTube data, please avoid using YouTube in the approach section. It can only be viewed as a potential.
> > >
> > > 2. `Missing details for Re-targeting Wrist Pose`: thanks for your replay. It resolves my concern.
> > >
> > > 3. `Missing details about transforms`: It is much clearer after your reply. It is also recommended to mentioned these details in the pdf. It is really helpful for readers to understand how VideoDex training data is generated.
> > >
> > > 4.  `method does not consider the surroundings of the human hand`: I am still not clear how the author combine these visual and action priors from passive data with the *physical constraints of how robots should move in the world*. I still do not understand why the method can deal with physical constraints. As the author mentioned in the reply, *This is reasonable as we are not relying on robot to exactly mimic the human but get a general action prior.* Then how can you model physical constraints of robot motion.
> > >
> > > 5.  `hand wrist pose and robot arm should also be visualized in Figure 3`: Thanks for making the new figure. But I do not understand that why the hand and robot arm can not be visualize in the same images. It is very hard to tell how large the difference is between original human motion in the image and the computed robot motion. This question is also related to the previous question. I recommend the author to visualize the robot arm and hand pose jointly and overlay it on the original RGB image to better show that whether it resemble the original human. I believe that how to generate training data from human video is key contribution of this paper but currently there is no visualization or numbers show the quality of the generated data compared with the original motion.
> > >
> > > 6. `benefits of using dexterous hand for the task defined..compared to normal parallel gripper.` and `Missing citations`: thanks for your effort. I do not have any concerns regarding now.
> > >
> > > In summary, I still have some concerns regarding 4 and 5 after reading the reply. Also as reviewer 1pwQ mentioned, many of the issues are about writing. It will be better if the author want to resolve these issues in the pdf.

---

### Author Response · Authors · 2022-08-23
**[All Reviewers/AC] Camera accelerometer not needed; VideoDex works from just videos**

One major concern that everyone raised is that VideoDex requires camera gyro/accelerometer data. We would like to clarify that this decision was more a result of convenience than necessity because EpicKitchens already had the camera data available. However, our method does not rely on this camera data and can work purely from just videos. Below, we provide three different ways.

Interestingly, we found that ***after removing the reliance on camera data, our performance improved as it no longer suffers from estimation error in the camera’s gyro/accelerometer data.***

### VideoDex without camera gyro/accelerometer data

We define $M^{C_1}_{World}$ to be parameterized by the vector $\alpha$ containing both a roll and pitch term. In the submission, $\alpha$ was obtained from the camera’s gyro/accelerometer data, but we present three ways to estimate it:

1. **Surface Normal**: All we need is a rough correspondence between human pose to robot pose. One way is to look at the surfaces around the human and use their normals direction to get a camera-independent way to transfer pose information.
- We recover object segmentations for surfaces that are parallel to the floor such as tables, floors, counters, and similar synonyms using a state-of-the-art object detector [1]. We then compute a depth map from RGB frames only using Adabins [2].  This does not rely on the long-term contiguity of a video like most SLAM approaches. We then use depth map portions that correspond to the relevant objects and calculate a surface normal vector.
- Using this vector, we estimate $\alpha$ similarly to how it was estimated from the accelerometer vector to obtain $M^{C_1}_{World}$.  We then use this transformation in our pipeline to make the trajectory upright to match the robot rotation.

2. **Random**:  A simple baseline is to just randomize $\alpha$ in the range of 15-45 degrees, given the assumption that most egocentric data is collected with the camera facing in front. This is reasonable as we are not relying on robot to exactly mimic the human but get a general action prior.

3. **Fixed**:  We assume that $\alpha$ is [0,0]. This is reasonable as we are not relying on robot to exactly mimic the human but get a general action prior.

### Removing camera gyro/accelerometer data for SLAM
In monocular RGB SLAM without sensors, one cannot disambiguate a scale factor, $s$ on the translation component of the camera movement.  Originally, we used the gyroscope and accelerometer data to determine $s$ for SLAM.  With this change, we do not use gyro and accelerometer data and use a fixed scale factor of 1.0. The scale factor does not need to be accurate. This is because in the final stage,  we center and scale the trajectory down to ensure it fits in the robot frame anyway.

### Wrist Retargeting Clarification
The wrist-retargeting does **not need any extra information other than the 2D image**. The FrankMocap [39] hand model detects paired 2D keypoints in the image and 3D keypoints in the SMPLX [37] canonical frame. We use the Perspective-n-Point approach to transform these 3D keypoints into the camera frame. COLMAP [65] is used (which only needs 2D images) to estimate camera intrinsics. All of the steps of our updated retargeting approach are obtainable from generic human videos without any additional data.

### Results
We present the results of the previously discussed experiments **without using any extra camera data** on the place, cover and uncover tasks.

1. **VideoDex-Surface**: VideoDex with the angle being output from the surface normal estimation using an object detector to find a flat surface such as the floor or table.
2. **VideoDex-Random**: VideoDex with the angle being sampled randomly in the range of 15-45 degrees (with respect to the floor), since the camera is likely facing the front.
3. **VideoDex-Fixed**: VideoDex that assumes no transformation between the human and the robot frame.

|Task/Method | VideoDex-Original | VideoDex-Fixed | VideoDex-Random | VideoDex-Surface|
|--------------|-------------------|----------------|-----------------|-----------------|
| Place         | 0.70      |0.55      |0.45       |**0.80**|
| Cover        | **0.67**      |0.45      |0.63       |0.63|
| Uncover    | 0.90      |0.77      |0.85       | **0.92**|

- We see that performance degrades when randomizing or setting $M^{C_1}_{World}$ to a fixed value, in all three of the tasks, but it is still comparable to or better than our baselines that do not use any human action data.
- More interestingly, we found that $\texttt{VideoDex-Surface}$ performed better than our original VideoDex without using any camera sensor data. A possible explanation for this is that the sensor data may be noisy and estimating surface normals from visual features may be more robust. Note that all of the reported numbers are success rates on test objects.

Hope we addressed this major concern. For other concerns, please refer to individual answers.

---

### Meta-Review · Area_Chair_Hub2 · 2022-08-06

**Recommendation:** Accept (Poster)
**Confidence:** 4

**Metareview:**

This paper studies how to learn dexterous manipulation from human videos.

In the initial review, the reviewer appreciated the direction and real-world experiment but also raised  concerns about the need of special sensor for tracking. During rebuttal, the authors effectively addressed this concern by providing additional experiment results, and reviewers were satisfied with the response.
AC would like to recommend acceptance for this paper.

**Best Paper Nomination:**

No

---

> ### Author Response · Authors · 2022-08-23
> **Response to AC # Hub2**
>
> We thank the reviewers and Area Chair for their valuable feedback and have addressed these in individual responses as well as below.
>
> > *Overclaim learning from youtube videos, while a special sensor for the camera is needed.*
> - In our submitted version, we use the camera’s gyro/accelerometer data to transform the human camera pose to that of the robot's. This was more of a convenience than necessity as Epickitchen already had the camera data. However, our method does not rely on this camera data and can work purely from just videos.
> - Interestingly, we found that after removing the dependency on camera gyro/accelerometer data, the performance of VideoDex improved as it does not suffer from estimation error in camera’s gyro/accelerometer data. Please see the results and approach in the common comment above, or click here: [https://openreview.net/forum?id=qUhkhHw8Dz&noteId=pGj0s01CD71](https://openreview.net/forum?id=qUhkhHw8Dz&noteId=pGj0s01CD71)
> - In addition, we have modified the paper to tone down all the claims significantly and reflect our findings as well as misunderstandings about the technical details.
>
> > *Missing literature discussion*
> - We add many relevant papers to further highlight the many areas of research we draw from.  We have added a section called “Robot Learning by Watching Humans”, which better highlights many key papers in this subfield’s history.
>
> > *Missing details*
> - We have clarified details regarding both the human action retargeting system as well as the experimental details. We respond to the reviewer’s questions and add information to the paper to clarify the different mechanisms in the retargeting system.  We clarify details including how to compute $M_{Wrist}^{C_t}$ from PnP, how to compute  $T_{World}^{Robot}$ and what data augmentations were used. **This is all done from 2D RGB images only.**
>
> *Additional Ablations*
> - Reviewers' also asked for multiple new ablations, specifically looking at how well the visual priors behave, if action priors can transfer from one task to the other, and investigating the sensitivity of VideoDex to noisy data. We are pleased to report that we finished running all of them and posted them below in individual replies.